# Optimization of Blood Handling and Peripheral Blood Mononuclear Cell Cryopreservation of Low Cell Number Samples

**DOI:** 10.3390/ijms22179129

**Published:** 2021-08-24

**Authors:** Christopher M. Hope, Dao Huynh, Ying Ying Wong, Helena Oakey, Griffith Boord Perkins, Trung Nguyen, Sabrina Binkowski, Minh Bui, Ace Y. L. Choo, Emily Gibson, Dexing Huang, Ki Wook Kim, Katrina Ngui, William D. Rawlinson, Timothy Sadlon, Jennifer J. Couper, Megan A. S. Penno, Simon C. Barry

**Affiliations:** 1Robinson Research Institute, Adelaide Medical School, University of Adelaide, Adelaide, SA 5005, Australia; christopher.hope@adelaide.edu.au (C.M.H.); dao.huynh@adelaide.edu.au (D.H.); yingying.wong@adelaide.edu.au (Y.Y.W.); helena.oakey@adelaide.edu.au (H.O.); griffith.perkins@adelaide.edu.au (G.B.P.); t.nguyen@adelaide.edu.au (T.N.); timothy.sadlon@adelaide.edu.au (T.S.); jennifer.couper@adelaide.edu.au (J.J.C.); megan.penno@adelaide.edu.au (M.A.S.P.); 2Women’s and Children’s Hospital, Adelaide, SA 5006, Australia; 3Children’s Diabetes Centre, Telethon Kids Institute, The University of Western Australia, Perth, WA 6009, Australia; Sabrina.Binkowski@telethonkids.org.au (S.B.); Ace.Choo@health.wa.gov.au (A.Y.L.C.); 4Child Health Research Unit, Barwon Health, Geelong, VIC 3220, Australia; minh.bui@barwonhealth.org.au; 5School of Women’s and Children’s Health, Faculty of Medicine and Health, University of New South Wales, Sydney, NSW 2052, Australia; Emily.Gibson@health.nsw.gov.au (E.G.); k.w.kim@unsw.edu.au (K.W.K.); w.rawlinson@unsw.edu.au (W.D.R.); 6Walter and Eliza Hall Institute of Medical Research, Melbourne, VIC 3052, Australia; dhuang@wehi.edu.au (D.H.); ngui@wehi.edu.au (K.N.); 7Virology Research Laboratory, Serology and Virology Division, NSW Health Pathology, Prince of Wales Hospital, Sydney, NSW 2031, Australia

**Keywords:** PBMC, blood handling, delay in processing, cryopreservation, cell concentration

## Abstract

Background: Rural/remote blood collection can cause delays in processing, reducing PBMC number, viability, cell composition and function. To mitigate these impacts, blood was stored at 4 °C prior to processing. Viable cell number, viability, immune phenotype, and Interferon-γ (IFN-γ) release were measured. Furthermore, the lowest protective volume of cryopreservation media and cell concentration was investigated. Methods: Blood from 10 individuals was stored for up to 10 days. Flow cytometry and IFN-γ ELISPOT were used to measure immune phenotype and function on thawed PBMC. Additionally, PBMC were cryopreserved in volumes ranging from 500 µL to 25 µL and concentration from 10 × 10^6^ cells/mL to 1.67 × 10^6^ cells/mL. Results: PBMC viability and viable cell number significantly reduced over time compared with samples processed immediately, except when stored for 24 h at RT. Monocytes and NK cells significantly reduced over time regardless of storage temperature. Samples with >24 h of RT storage had an increased proportion in Low-Density Neutrophils and T cells compared with samples stored at 4 °C. IFN-γ release was reduced after 24 h of storage, however not in samples stored at 4 °C for >24 h. The lowest protective volume identified was 150 µL with the lowest density of 6.67 × 10^6^ cells/mL. Conclusion: A sample delay of 24 h at RT does not impact the viability and total viable cell numbers. When long-term delays exist (>4 d) total viable cell number and cell viability losses are reduced in samples stored at 4 °C. Immune phenotype and function are slightly altered after 24 h of storage, further impacts of storage are reduced in samples stored at 4 °C.

## 1. Introduction

The components of peripheral venous blood can provide a minimally invasive bioresource to aid the identification, understanding, and treatment of many diseases [1,2]. Peripheral blood mononuclear cells (PBMC) are a valuable resource for a wide range of immunological studies from use as feeder cells in cell cultures, for immune cell profiling and functional characterization, as well as the starting material for a genetically modified cellular therapy. The isolation, cryopreservation, and thawing of PBMC from adult blood have been extensively investigated with several previously published methodological papers [3,4,5]. 

The most widely utilized method for obtaining PBMC is via density gradients or barriers that allow granulocytes, red blood cells (RBC) and RBC aggregates to pass through while lymphocytes, monocytes and platelets are maintained in a separate layer [6,7,8]. The same isolation procedures are used in cord blood and pediatric blood samples [9,10]. 

Previous reviews suggested several factors that contribute to the isolation of high quality and high yield of PBMC [11,12]. Assuming adequate staff training and a satisfactory isolation method, for example, using methods accredited by Biospecimen Proficiency Testing in Integrated Biobank of Luxembourg (IBBL) [13]; the major factors that influence PBMC quality are (i) delays to processing, (ii) the volume of blood drawn, and (iii) the temperature which the blood is stored at prior to processing. Reductions in PBMC count per milliliter of blood were observed as early as 8 hours post-venipuncture [14,15] and PBMC losses are evident at 24 h [16]. Several reports indicated delaying blood processing after venipuncture also causes a reduced functional capacity of different cell types. For example, a significant loss (36–56%) in IFN-γ T cell frequencies by enzyme-linked immune absorbent spot (ELISPOT) assay was observed when the blood was processed 24 h after venipuncture. These losses are coupled with increases in low density activated granulocytes, i.e., mainly neutrophils [17]. Therefore, almost all studies to date recommend a short timeframe (<8 h) between blood venipuncture and PBMC isolation. However, this may be impossible to achieve in multi-center clinical trials that cover a large geographic region, for example, the Regional Participation Program arm of the Environmental Determinants of Islet Autoimmunity (ENDIA) study in Australia [18]. In the effort to mitigate the adverse effect of delay to processing on cell loss, thus increasing the quality of PBMC collected in such trials, Olson et al. [19] found that temperature at which the blood is stored prior to processing impacted cell recovery and viability and suggested that ambient temperature (above 22 °C or preferably near 30 °C) protected the cells during overnight shipments. 

When smaller volumes of blood are drawn, fewer PBMC are obtained. A reduced cell number will reduce the cryopreservation media volume and/or decrease the PBMC concentration at LN2 storage. The issue becomes more important in studies involving pediatric populations, as drawing large amounts of blood from infants has practical and ethical constraints. Furthermore, the cryopreservation process itself also contributes to cell loss due to the use of cryoprotectants, e.g., dimethyl sulfoxide (DMSO) [20,21]. To our knowledge, there are no studies that have been conducted to explore a low volume cut-off, where the PBMC are stored without affecting cell viability and recovery post-thaw. Whilst there have been efforts to alter the PBMC concentration, these studies increased the concentration of PBMC to decrease the volume and the number of vials required to store large amounts of PBMC. It was found that PBMC can be stored at concentrations as high as 3 × 10^7^ cells/mL [11,22]. However, the impact of reducing PBMC concentrations below 5 × 10^6^ cells/mL is rarely mentioned [12]. No studies have explored whether any viable cells can be recovered after thawing PBMC at concentrations lower than 5 × 10^6^ cells/mL, especially in low volume cryoprotectants, as found when storing blood samples that may have been drawn from a child.

This study investigated the optimal conditions at which blood should be stored in case of unavoidable extended delays to processing and aimed to determine the feasibility of storing low-volume blood samples. We also reveal optimized storage conditions for PBMC yields as low as 1 × 10^6^ cells at which post-thaw viability and recovery were equal to cells stored at 5 × 10^6^ cells in 500 µL. These optimization steps will enable efficient collection and recovery for the expected PBMC yields that might be isolated in 0.5–1 mL of pediatric blood or those study participants with a limited blood draw. 

## 2. Results

### 2.1. Experiment 1: Effects of Storage Temperature and of Delayed Processing on PBMC Viability and Recovery

#### 2.1.1. Storage Assessment

We first measured the impact of delayed processing (daily, for 10 days) on PBMC yield and viability and whether storing the blood at 4 °C could mitigate cell losses over time. Test samples were collected from 10 healthy adult donors and stored at two temperatures (n = 5 at room temperature (RT), n = 5 at 4 °C) prior to their isolation and preparation for cryopreservation at a cell density of 10 × 10^6^ cells/mL. The changes of total viable cell number and viability over 10 days of blood storage are shown (Figure 1).

The results of the linear mixed model showed that both storage temperature and the number of days stored prior to processing, and their interaction, significantly impacted viability (*p* < 0.001) and total viable cell number (*p* < 0.001) at the time of cryopreservation. The number of days in which blood was stored impacted total viable cell number and viability when stored for one day at 4 °C and two days at RT (Table 1). However, significantly higher total viable cell numbers and higher viability were found in 4 °C samples compared with RT samples when the blood was stored for 4 or more days prior to processing (Figure 1). 

Significant changes in total viable cell number occurred after one day of storage at 4 °C with 0.39 × 10^7^ fewer cells whereas RT had 0.29 × 10^7^ fewer cells (Table 1). The viable cell number decreased rapidly after 4 days at RT down to 0.04 × 10^7^ (95%CI: 0.00 × 10^7^, 0.11 × 10^7^) with a loss of 0.85 × 10^7^ cells after 3 days, whereas 4 °C had a decrease of 0.32 × 10^7^ (95%CI: 0.06 × 10^7^, 0.54 × 10^7^) being significantly higher than RT (*p* < 0.05, Table 1).

Viability was 18.9% lower after one day at 4 °C, whereas the viability at RT after one day had decreased by just 7.3%, with significant differences from initial viability coming two days after storage in RT (Table 1). However, viability steeply declined in RT, when samples were stored 4 days or more, the viability was significantly lower (*p* < 0.001) in RT than those stored at 4 °C, with viability at 8.9% (95%CI: 3.4, 18.4) in RT samples after 4 days compared with 54.3% (95%CI: 13.2, 77.5) in those at 4 °C (Figure 1).

#### 2.1.2. Post-Thaw and Post-Incubation Assessment

A cryovial of each sample was removed from liquid nitrogen storage and used for post-thaw and post-incubation. The number of days delayed before processing significantly affected both the viability and recovery of cells post-thaw and post-incubation (*p* < 0.001). Interestingly, blood storage temperature did not impact significantly on variables measured after cryopreservation (*p* > 0.05). 

Predicted means of viability and recovery of samples post-thaw and post-incubation by storage day are shown in Figure 2a and Figure 2b respectively.

Samples stored and processed after 2 days of delay were 35.2% lower in post-thaw viability (Figure 2a) compared with samples freshly processed (day 0). Furthermore, no significant difference in post-thaw recovery was found in samples between processing (day 0) and day 2. Post-incubation viability measured samples at day 1 were 34.7% lower than that of samples immediately processed (Figure 2b) and recovery had reduced by 29.7% at day 1. When data from samples measured at the time of processing and four days later are compared, all post-thaw and post-incubation measures, i.e., viability and recovery at day 4 were significantly lower than those at day 0 (*p* < 0.05, Figure 2).

#### 2.1.3. Cell Functionality

Cell function and immune phenotype (2.1.4) were performed on a cell-limited sub-cohort. We had 23/30 (76.7%) of samples stored; Day 0 (90%), Day 1 (100%) and Day 2 (40%). A further breakdown shows we had 11/15 (73%) of samples stored at RT and 12/15 (80%) of samples stored at 4 °C. The major limitation to this sub-study is we only had one sample 1/5 (20%) which was stored at RT for 2 days.

Blood storage temperature, number of storage days prior to processing and their interaction did not affect CEF (Cytomegalo, Epstein–Barr and Flu virus) peptide stimulated IFN-γ ELISPOT counts (*p* > 0.05). Of note, the impact of storage days prior to processing *p*-value was equal to 0.068, due to the limited number of RT day 2 samples (n = 1).

There was no significant difference of (Phytohemagglutinin) PHA stimulated IFN-γ ELISPOT counts between 2-day old samples compared with samples processed immediately (*p* > 0.05). However, the count significantly reduced by 933 spots (~70%) when the samples were stored 24 h prior to processing, regardless of storage temperature (Appendix A). 

#### 2.1.4. Immunophenotyping of Cell Population

*Low-Density Neutrophils.* Blood storage temperature, number of storage days, and their interaction, significantly impacted low-density neutrophil percentage. No significant differences were found between samples stored at 4 °C compared with samples stored at RT, at day 0 and day 1. However, the neutrophil percentage of RT samples dramatically increased from 3.7% (95% CI: 0.9, 6.6) at day 1 to 22% (95% CI: 17.9, 26.4) at day 2 which is significantly higher than 4 °C at day 2 (4.4% (95% CI: 1.3, 7.6), *p* < 0.001) (Appendix A). 

*Monocytes*. The number of storage days prior to processing decreased the monocytes population (*p* < 0.05). Monocytes significantly decreased by 8.3% after one day of storage (*p* < 0.05) with no difference at day 2 (Appendix A). 

*Lymphocytes.* There was no impact of blood storage temperature on lymphocytes percentage. However, the percentage of lymphocytes significantly increased from 84.3% (95%CI: 79.8, 87.9) to 88.6% (95%CI: 85.2, 91.2) after one day of storage, but then decreased significantly to 79.5% (95%CI: 71.4, 85.7) at day 2 compared with day 1. Day 2 was not significantly different from day 0 (Appendix A). 

*T-cells.* The T-cell percentage was significantly affected by both the temperature and storage day number (interaction, *p* < 0.05). There were no significant differences in T-cell percentage between 4 °C samples and RT samples at day 0 and day 1 of storage. However, RT samples showed a 16% significantly higher T-cell percentage than 4 °C samples by day 2 (Appendix A). Furthermore, blood storage temperature and the number of days in which blood was stored prior to processing did not impact the CD4 and CD8 populations (*p* > 0.05). 

*NK cells.* A significant decrease of 25% of NK (CD56+CD16+) cells was measured in samples stored for 1 day, compared to samples processed on day 0 (*p* < 0.05). This percentage continued to drop to 12.5% (95%CI: 0, 30.3) when the samples were processed after 2 days of storage (Appendix A).

*B cells.* Both blood storage temperature and the number of days which blood was stored prior to processing impacted the B cell percentage, (interaction, *p* < 0.05). The 4 °C samples had a higher mean percentage of [6.9% (95%CI: 4, 10)], compared to RT samples [2.3% (95%CI: 0, 6.3)] at day 3. 

### 2.2. Experiment 2: Impact of Cryopreserved Cell Volume on PBMC Quality Evaluated Post-Thaw and Post-Incubation

In experiment 1, the total viable cell number at the time of cryopreservation significantly predicted post-thaw viability after adjustment for the storage temperature and the number of days which blood was stored prior to processing (2.5 (95%CI: 0.57, 4.43), *p* < 0.05)). When keeping cell concentration constant, alterations to total cell number alters the volume those cells are resuspended in, i.e., a lower amount of PBMC will be in a lower volume of cryoprotective media. 

To investigate the protective effect of low (<500 µL) cryoprotective media volume, we first determined the lowest volume of cryoprotective media that could preserve PBMC without compromising viability and recovery based on acceptable criteria: post-thaw viability ≥80% and recovery of ≥55% [23]. Samples were obtained from three healthy adult donors with PBMC viability >90% at collection and cryopreserved at seven different volumes (500, 300, 200, 150, 100, 50, and 25 µL) each at a PBMC concentration of 10 × 10^6^ cells/mL. Thawing was performed following storage in liquid nitrogen for an average of 4.4 months. 

The mean viability and recovery percentages measured post-thaw and post-incubation for each cryopreservation volume are shown (Figure 3a). Post-thaw viability and recovery means were within the acceptable criteria range for all sample volumes (Figure 3b). The cryopreserved volume impacted the post-thaw viability (*p* < 0.001). The Tukey’s test showed no significant difference in post-thaw viability between samples cryopreserved at volumes ≥100 µL. The viability of samples cryopreserved at 50 µL (83.1, 95%CI: 79.7, 86.4) and 25 µL (82.1, 95%CI: 79.8, 85.5) was significantly less than that of samples cryopreserved in volumes ≥200 µL (*p* < 0.05, Figure 3b), but no significant differences between samples cryopreserved at these volumes at either 100 µL or 150 µL. These volumes were the only ones where the 95% confidence intervals dropped just below the acceptable criteria. No significant differences were observed between volumes in post-thaw recovery (*p* > 0.05).

The mean post-incubation viability of cells cryopreserved at different volumes ranged from 73.7% to 90.3% and the mean recovery ranged from 32.9% to 57.8% (Figure 3c). The cryopreservation volume significantly impacted the post-incubation PBMC viability (*p* < 0.001) and post-incubation recovery (*p* < 0.001). Lower viability measured post-incubation was found in samples cryopreserved at either 25 µL or 50 µL, compared to samples cryopreserved at ≥150 µL. Although samples cryopreserved at 150 µL had no significant differences in recovery or viability post-thaw from samples cryopreserved at 100 µL (*p* > 0.05), the viability achieved in 150 µL samples (88.9%, 95%CI: 86.0, 91.7) was approximately 6% higher than that in 100 µL samples (83.4%, 95% CI: 80.2, 86.3). For post-incubation recovery, no significant differences were found between 150 µL samples and ≥200 µL sample, while 100 µL samples (42.3%, 95% CI: 37.9, 46.7) exhibited significantly lower post-incubation recovery than ≥200 µL samples (*p* < 0.05). This was considered a biologically significant decrease, indicating that 100 µL is not sufficient to protect PBMC from dying after cryopreservation and an overnight incubation at that cell concentration. 

### 2.3. Experiment 3: Impacts of Cryopreserved Cell Concentration on PBMC Quality Evaluated Post-Thaw and Post-Incubation

To dissect whether the storage volume of cryopreservation media or overall cell number was providing cryoprotection, using PBMC from four healthy adult donors, the media volume was kept constant at 150 µL with four different total cell concentrations: 10 × 10^6^, 6.67 × 10^6^, 3.33 × 10^6^ and 1.67 × 10^6^ cells/mL (total cell numbers: 1.5 × 10^6^, 1 × 10^6^, 0.5 × 10^6^, and 0.25 × 10^6^ per vials). Of note, the lowest concentration used was the same number of cells stored in the lowest volume of experiment 2 (i.e., 25 µL). The mean viability and recovery percentages post-thaw and post-overnight incubation stored at different cell concentrations are shown in Figure 4a. Post-thaw viability and recovery of samples cryopreserved at 6.67 × 10^6^ cells/mL and 10 × 10^6^ cells/mL achieved the acceptable criteria. Although the viabilities of these samples were still over 80% after incubation, the recovery dropped significantly below 40% for all cell concentrations. 

The cell concentration at cryopreservation statistically impacted the viability of cells after thaw (*p* < 0.01, Figure 4b), while no effect of cell concentration was observed on post-thaw recovery (*p* > 0.05). Significant differences in post-thaw viability were found between either sample cryopreserved at 10 × 10^6^ or 6.67 × 10^6^ cells/mL compared with samples stored at 1.67 × 10^6^ cells/mL. No differences were found in other pairs of samples. Following overnight incubation at 37 °C, there was significantly higher viability found in samples stored at 10 × 10^6^ cells/mL and 6.67 × 10^6^ cells/mL compared with 1.67 × 10^6^ cells/mL (Figure 4c). Again, cell concentration had no impact on the recovery of samples measured after overnight incubation (*p* > 0.05).

## 3. Discussion

The results of this study indicate that storing blood samples at RT for <24 h does not statistically alter viable cell count and viability when compared with processing without delay. However, it is beneficial to store samples at 4 °C, if it is known that the delay to processing will be >24 h post-venipuncture, to slow the inevitable sample degradation. Firstly, it was observed that 4 °C storage prior to processing had a greater number of viable cells after 3 days and higher viability after 4 days post-venipuncture compared with samples stored at RT for the same length of time. Secondly, after 24 h there were significant differences in immune phenotype between samples stored at RT, which had a larger proportion of Low Density (LD) neutrophil contamination and a slightly higher T cell proportion compared with samples stored at 4 °C. However, temperature could not mitigate the proportional loss of monocytes or NK cells. Lastly, samples stored with a delayed of 2 days at 4 °C did not have significantly different CEF or PHA stimulated IFN-γ release compared with day 0 baseline samples, however, samples stored for 1 day had a decreased PHA stimulated IFN-γ release. This study also investigated the lowest possible cryoprotectant and PBMC concentration without statistically altering post-thaw viability and viable cell number immediately after thawing and after overnight incubation. We observed that PBMC can be stored at the concentration of ≥6.67 × 10^6^ cells/mL in a sample volume of ≥150 µL before there were any statistically significant differences observed in viability and viable cell numbers post-thaw and post-incubation, maintaining cell integrity at the acceptable levels for downstream analysis.

PBMC are an integral part of many immunological research projects. Blood handling prior to PBMC processing, isolation method, and optimization at cryopreservation are discussed in many studies and reviews [4,11,19,24,25]. Two circumstances that impact yield and quality, specifically extended delay of processing and low volume of blood drawn from donors, have not been investigated as extensively. A study functionally tested PBMC acquired from a time-course in delaying PBMC processing showed that a short delay to processing (<8 h) impacted the integrity of cells [15]. However, an 8-h maximum delay is often exceeded in multi-site clinical trials, as blood samples may have to be transported from rural or remote sites to well-equipped central laboratories for PBMC isolation and cryopreservation. Attempting to reduce the impacts of the delays in processing on PBMC quality due to transportation, one multi-center study transported blood to a central laboratory at a variety of transportation temperatures. The study found that blood samples exposed to 15 °C or 40 °C had lower quality compared with samples stored at 22 °C and 30 °C [19]. This was further supported in another study, showing ambient temperature shipping of blood did not impact post-thaw viability and recovery [26]. Our results confirm that when storing blood for 24 h prior to PBMC isolation, RT storage did not have a statistically significant impact on the viability and total viable cell number whereas 4 °C had significantly lower viability and viable cells. Furthermore, blood samples stored for >24 h had a continuous decrease in total viable cells measured over 4 days of blood storage, independent of blood storage temperature. However, within the same timeframe, the viability in samples stored at RT continuously reduced, while those stored at 4 °C only decreased for the first 2 days and maintained a >40% viability until day 9 of storage. 

Many clinical studies do not use PBMC fresh, we investigated how blood storage temperature and delay to processing impacted the quantity and quality of cryopreserved PBMC. Lowering blood storage temperature during delay did not significantly change post-thaw viability or total viable cells recovered. Furthermore, in samples stored >2 days, we observed a stepwise decrease in post-thaw viability, however, a 3-day plateau or delay in reductions of post-thaw recovery. It is noted that many studies include an incubation period (up to 24 h) to remove any cells that are undergoing apoptosis due to the thawing process [27]. We incubated thawed PBMC for 16–18 h, post-incubation viability and recovery were also not impacted by storage temperature. However, it is evident that after 24 h (1 day) delay, there is lower viability and recovery, interestingly this impact is less pronounced in samples stored for 2 days. Furthermore, post-incubation viability is similar across days 1, 2 and 3. Although statistically not significant, samples stored at 4 °C trended towards having higher post-thaw recovery and post-thaw viability ≥2 days of storage compared with samples stored at RT. Samples stored at 4 °C for 2 days had a mean recovery of 78% versus 50% and day 2 mean viability of 68% versus 35%, respectively. Furthermore, blood stored at 4 °C for 3 days had a post-thaw recovery of 44% and post-thaw viability of 55% indicating the potential use of cells prior to an overnight incubation period. Viable cell numbers and viability does not provide the cell composition or functional changes that these PBMC may have undergone. We investigated changes in immune phenotype and IFN-γ release using flow cytometry and ELISPOT, respectively.

The immune phenotype of the thawed PBMC revealed a decrease in both the monocyte and NK cell populations within the first 24-h time delay to processing. This observed decrease was not mitigated or enhanced by storing the blood at 4 °C prior to processing. These results are in line with others who have observed that storing blood samples after the collection had adverse effects on the monocyte [28] and NK [27] populations. Lymphocytes, defined by physical parameters only (i.e., FSC and SSC), had an increase of 5% on day 1, which may be purely due to the decrease in reciprocal cell types defined physically, although others have published similar findings [15,29,30,31,32]. Furthermore, we observed an increase in LD neutrophil and CD3+ T cell populations at Day 2 in thawed PBMC from the blood that was stored at RT prior to processing when compared with samples stored at 4 °C prior to processing. It was previously published that aged blood has higher neutrophil contamination compared to blood freshly processed [17,33] Our findings report that CD4 T cell and CD8 T cell proportions remained unchanged after 2 days of blood storage prior to processing. Samples stored >24 h at RT had a decrease in B cells to 2.3% (95%CI: 0, 6.3) by day 2. These findings were in line with previous literature showing that T and B cells were the most stable populations [28].

The thawed PBMC showed some differences in immune phenotype however it was warranted to investigate PBMC function. Peptide pool CEF and mitogen PHA were used to stimulate PBMC and IFN-γ release was measured using an ELISPOT assay. The linear regression model indicates that there was no significant difference in CEF stimulated IFN-γ release from samples with no delay, compared with 1 and 2 days of delay of processing (*p* = 0.07). 

This observation of a “recovery” of the sample’s immune phenotype and function after the initial 1-day of delay needs to be further investigated. We can only speculate that the initial blood draw contains or initiates an innate inflammatory response (perhaps by the monocytes and NK cells) which leads to an anti-inflammatory response within 24 h and apoptosis of these cells occurs, which reduces these cell populations. After 2 days of 4 °C storage, and a freeze-thaw cycle, the surviving T cells begin to respond to CEF and PHA to a similar level as they may have without any delay in processing, in absence of monocytes and NK cells. It is within this time (2 days) the neutrophils in the blood samples stored at RT are also activating, potentially by monocyte/NK cell apoptosis, and becoming LD neutrophils which can be harvested within the PBMC layer. It is perhaps these series of events that cause RT samples to be restricted in CEF and PHA stimulation and simply placing them at a lower temperature stall or stops these events from happening, at least within 2 days of storage.

No studies to date explored the quality of PBMC when blood was stored beyond 24 h and whether the storage temperature of blood prior to processing impacts the integrity of PBMC isolated from these samples. Careful consideration into the collection of blood and isolation of PBMC needs to be given. Selecting blood storage temperature, i.e., RT for processing within a 24 h time period or 4 °C when the time interval exceeds 24 h. For storing blood >24 h we and others recommended 4 °C as it provides more stability in multiple parameters [34] Furthermore, 4°C storage is also beneficial to other blood components which can be separated before PBMC processing, e.g., plasma [34]. Our findings provide valuable information for establishing Standard Operating Procedures (SOPs) around the collection and transport of blood collected at rural/remote sites and shipped to central laboratories more than two days post-venipuncture. This methodology is being applied to the Regional Participating Program of the ENDIA study [18]. 

Furthermore, no studies to date have explored differences in viability and recovery of samples cryopreserved at low (<500 µL) cryopreservation volumes and low (<5 × 10^6^ cells/mL) cell concentrations. This issue becomes more apparent in studies involving pediatric samples due to the clinical and ethical restrictions associated with blood draw volumes [35]. We explored the minimum volume that PBMC could be cryopreserved in to achieve an acceptable level of viability and recovery post-thaw and post-incubation. Our study is the first study reporting that post-thaw viability and recovery of samples cryopreserved at ≥150 µL passed the acceptable criteria (i.e., post-thaw viability ≥80% and post-thaw recovery ≥55%) as suggested by the Centre for HIV/AIDS Vaccine Immunology (CHAVI) [23]. 

We explored the lowest cell concentration that at 150 µL sample could still be protected without reductions in viability and recovery. We found that a cell concentration of 3.33 × 10^6^ cells/mL did not experience a statistically significant drop in viability or recovery post-thaw or after overnight incubation. However, these samples dropped below the acceptable CHAVI criteria for viability. A cell concentration of 6.67 × 10^6^ shows viability values within the CHAVI criteria and would be our recommendation for cryopreservation media volumes of 150 µL. There were several studies conducted to explore the impacts of cell concentration at the time of cryopreservation on the viability and recovery of PBMC measured directly after the thaw. One of the major differences between the published study and our current investigation is using a fixed storage volume of 1000 µL and lowering volumes was not previously evaluated. The results of previous studies reported no differences in post-thaw viability between samples stored at different cell concentrations, however, no measurements were performed after incubation [12,22]. In summary, our results suggest that low cell number (1 × 10^6^ or more) PBMCs should be cryopreserved at a cell concentration of ≥6.67 × 106 cells/mL at a sample volume of ≥150 µL, to ensure a high percentage of viability and recovery after thaw and after overnight incubation.

A limitation to this study is the small numbers processed for immune phenotyping and functional analysis, especially the single sample as a representation for samples stored at RT for 2 days. These techniques utilize quite a large number of PBMC, with study participants not having enough material to store across multiple vials. The immune phenotype and function were performed later than the initial post-thaw and post-incubation experiments, which meant material may not have been available for phenotyping and functional assays. Limitations of experiments 2 and 3 of this study include: using adult blood, however, the results may be extrapolated to pediatric samples, another was not including a cell concentration of 5 × 10^6^ cells/mL, however, it may be speculated from this study and others to not be impacted, with the caveat that it may be used with caution when storing samples in a volume of 150 µL.

This study is the first that we know of that has investigated decreasing the number of PBMC cryopreserved to levels equivalent to those expected from pediatric or neonate samples. 

## 4. Materials and Methods

### 4.1. Experiment 1: Effects of Storage Temperature and Delayed Processing on PBMC Viability and Recovery

Design: Ten healthy adult donors donated 60 mL of blood (10 × 6 mL) tubes containing ethylenediaminetetraacetic acid (EDTA) anticoagulant. Samples were non-identifiable and no personal information was collected as part of the study. Research was approved by Women’s and Children’s Health Network-Human Research Ethics Committee (WCHN-HREC). Approval Code: HREC/19/WCHN/65. Approval Date: 11 June 2019.

Fifty tubes were placed at RT and 50 tubes stored in a temperature-monitored fridge at 4 °C (Appendix A). One tube of each donor was routinely processed to isolate PBMC from day 0 to day 9 as described below. A temperature data logger was placed in the laboratory to investigate changes in room temperature over 24 h, and no significant differences were observed across experiments. The range of temperature was 22–27 °C. 

Isolation and cryopreservation of PBMC: Following the experimental time course described above, the whole blood EDTA tubes were centrifuged at 1700× *g* for 10 min. The buffy coat layer was harvested and placed into a 10 mL centrifuge tube diluted up to 7 mL with wash solution (1% Cosmic Calf Serum, CCS (GE Healthcare Bio-Science, Linz, Austria) in PBS (Sigma-Aldrich Pty. Ltd, Sydney, Australia). Three milliliters of Ficoll Paque^TM^ plus (GE Healthcare, Uppsala, Sweden) was underlaid and tubes centrifuged at 1000× *g* for 20 min at RT for the blood on Day 0 and for 30 min due to neutrophil contamination for all other blood on Day 1 onwards. The PBMC monolayer was harvested and re-suspended in 10 mL of wash solution and centrifuged at 480× *g* for 10 min. RBC lysis was performed on samples stored beyond two days by using lysis buffer consisting of NH_4_Cl (1550 mMol/L), KHCO_3_ (100 mMol/L and EDTA (1mMol/L) (produced by Walter and Eliza Hall Institute of Medical Research, Victoria, Australia). PBMCs were resuspended in 10 mL and counted on a Countess II cell counter (ThermoFisher, Cambridge, MA, USA). 

PBMC counting: 10 µL of the sample was added to 10 µL Trypan Blue (Sigma-Aldrich Pty. Ltd, Sydney, Australia) with 10 µL of the mixture added to a reusable glass slide of a Countess II. A size setting of 3–15 was the optimal count setting. Six readings (counts and viability) were taken within 2 min, approximately every 25 s. The first reading was considered an outlier and was removed; the average is found across 3/5 consistent readings.

Cryopreservation: PBMC were then centrifuged at 300× *g* for 10 min and re-suspended at a viable cell concentration of 2 × 10^7^ cells/mL in CCS (unless experimentally altered). Equal volume of pre-mixed and cooled 20% DMSO/CCS was added dropwise. The final mixture was equally transferred to 2 mL Nunc cryovials (Greiner Bio-One, Frickenhausen, Germany) according to the resuspension volume. Vials were placed into a −1 °C/min CoolCell^®^ device (Biocision, Mill Valley, CA, USA) and stored in a −80 °C freezer overnight. All samples were transferred into liquid nitrogen the following morning. Donated Australian Red Cross Lifeblood (ARCLB) buffy coats utilized the same centrifugation speeds and time, however, volumes were scaled 5-fold. This method was accredited by IBBL Proficiency Testing [13].

Thawing PBMC: A completed X-Vivo culture media (LONZA, Walkersville, MD, USA) supplemented with 5% human serum (Sigma-Aldrich Pty. Ltd, Sydney, Australia), 2% of Hepes 1M (Gibco by Life Technology), 1% of L-Glutamin (Sigma-Aldrich Pty. Ltd., Sydney, Australia) was made and warmed up by water bath at 37 °C before the thaw was performed. A cryovial of each sample was removed from liquid nitrogen and placed on dry ice. Samples were thawed in a 37 °C water bath with an incubation time equal to 10 min/mL of cryopreservation media. Thawed and warmed cells were transferred to a fresh 10 mL tube. A volume of warm completed X-Vivo culture media supplemented with 0.04% of DNAse (Worthington Biochemical Corporation, Lakewood, NJ, USA), was added dropwise to the cryovial to end up 2mls to wash residual cells out. Eight milliliters of the warmed media was added to the sample dropwise at a rate of ≤1 mL/5 s. The tubes were slowly inverted and centrifuged at 500× *g* for 10 min. The supernatant was aspirated and the cell pellet was washed twice in warmed culture media without DNAse supplementation before suspension. The thawed PBMC was suspended at ~3 × 10^6^ cells/mL suggested by Ramachandran et al. [36]. Suspended cells were then counted on a ThermoFisher Countess II (method described above) to measure post-thaw viable cell number and viability, which was used to calculate the post-thaw recovery. 

The cells were then plated in 2 different types of cell culture plates (Thermo Scientific, Roskilde, Denmark), depending on the suspension volume of thawed cells. Less than or equal to 1ml per well was added in a 48-well plate when the suspension volume was >200 µL while the whole volume was added into one well of a u-bottom 96-well plate in case of suspension volume ≤200 µL. The empty surrounding wells were filled with PBS. The plates were left in a culture incubator at 56 °C from 16 to 18 h before being washed and then counted again. This PBMC thaw method was modified from the original method optimized by Ramachandran et al. [36]. The viability and viable cell number were measured to calculate the recovery (%) after overnight incubation as follows:(1)$$\mathrm{Recovery}\;\left(\%\right)\;\mathrm{direct}\;\mathrm{after}\;\mathrm{thawing}=\frac{\mathrm{number}\;\mathrm{of}\;\mathrm{viable}\;\mathrm{PBMC}\;\mathrm{after}\;\mathrm{thawing}}{\mathrm{number}\;\mathrm{of}\;\mathrm{viable}\;\mathrm{PBMC}\;\mathrm{at}\;\mathrm{storage}}\times 100$$
(2)$$\begin{array}{c} \mathrm{Recovery}\;\left(\%\right)\;\mathrm{after}\;\mathrm{overnight}\;\mathrm{culture}\\ =\frac{\mathrm{number}\;\mathrm{of}\;\mathrm{viable}\;\mathrm{PBMC}\;\mathrm{after}\;\mathrm{overnight}\;\mathrm{culture}}{\mathrm{number}\;\mathrm{of}\;\mathrm{viable}\;\mathrm{PBMC}\;\mathrm{at}\;\mathrm{storage}\;-\;\mathrm{number}\;\mathrm{of}\;\mathrm{PBMC}\;\mathrm{removed}\;\mathrm{for}\;\mathrm{measurement}\;\mathrm{directly}\;\mathrm{after}\;\mathrm{thawing}}\times 100 \end{array}$$

In some cases, the 10 µL aliquot for counting significantly reduced the cells number in the test sample, and this was adjusted for in the post-incubation count. Acceptable criteria for PBMCs suggested by CHAVI, specifically post-thaw viability ≥80% and post-thaw total cell recovery 55–120%, were used to evaluate the quality of PBMCs obtained in the experiment [23].

ELISPOT method: Millipore 96-well plates with nitrocellulose membranes (Merck, Branchburg, NJ, USA) were activated with 35% ethanol for 30 s, before washing twice with PBS. Wells were coated with anti-IFNγ capture antibody (Clone 2G1, ThermoFisher, Cambridge, MA, USA) overnight at 4 °C, then washed twice with PBS. PBMC were thawed and counted, then treated in duplicate with complete media (RPMI + 10% FCS, glutamate, penicillin, streptomycin) alone, and with PHA (7.5 μg/mL; Merck, Branchburg, NJ, USA) or a cytomegalovirus/Eppstein–Barr virus/influenza peptide pool (2 μg/mL of each peptide; MabTech) and incubated at 37 °C (5% CO_2_). Wells were prepared with treatment media at 2× concentration and warmed for 1 h at 37 °C prior to addition of the PBMC. After 16 h, wells were washed five times with PBS, then ten times with PBS + 0.05% tween-20. Captured IFNγ was detected with a biotinylated anti-IFNγ antibody (Clone B133.5; ThermoFisher, Cambridge, MA, USA) at 4 °C overnight. Unbound detection antibody was removed by washing with PBS + 0.05% tween-20, and a streptavidin: HRP conjugate (BD bioscience) was added for four hours at 4 °C. AEC substrate (BD biosciences) was added for 10 min at room temperature, before rinsing with deionized water and enumeration of spots using an ImmunoSpot analyzer and software (Cellular Technology Ltd., Bonn, Germany). All washing steps were performed using an automated plate washer.

Immunophenotyping method: Thawed and overnight rested PBMC were counted and plated 0.5 × 10^6^ cells per well in a 96 well plate. Cells were stained with cell surface markers; CD16-PE (neutrophils), CD14-FITC (monocytes), CD3-BUV737 (T cells), CD4-BUV496 (Helper T cells), CD8-BUV395 (Cytotoxic T cells), CD56-PerCP-Cy5.5 (NK cells), CD19-BV711 (B cells), for 20minutes, RT, in the dark. The cells were then washed and fixed in 0.4% PFA, glucose fixative for 20 mins, RT, in the dark. The fixed cells were then washed in 2% FBS/PBS and resuspended in 50 uL. PBMC samples were run on a BD FACS symphony and compensated in FCS express.

### 4.2. Experiment 2: Impact of Cryopreserved Cell Volume on PBMC Quality Evaluated Post-Thaw and Post-Incubation 

Peripheral blood buffy coats collected from four healthy adult donors were obtained from ARCLB (Material Supply Deed 19-03SA-02). Acceptable criteria, i.e., viability ≥80% suggested by Proficiency Testing Program in the IBBL, were used to select appropriate PBMC samples for cryopreservation after isolation. Three PBMC samples passed the acceptable criteria and PBMC viability ranged from 92–96% at cryopreservation. The samples were cryopreserved at seven different volumes; 500, 300, 200, 150, 100, 50 and 25 µL at a PBMC concentration of 10 × 10^6^ cells/mL The cells were stored in liquid nitrogen for an average of 4.4 months before thawing by using the method described above. Three individual vials were thawed from the 7 different volumes, i.e., the experiment was run in triplicate. Recovery post-thaw and overnight incubation were calculated. 

### 4.3. Experiment 3: Impacts of Cryopreserved Cell Concentration on PBMC Quality Evaluated Post-Thaw and Post-Incubation

Six healthy donor peripheral blood buffy coats were obtained from ARCLB. Four buffy coats passed acceptable criteria and PBMC viability was 91–97% at the time of cryopreservation. PBMC were isolated and cryopreserved at different cell concentrations: 10 × 10^6^, 6.67 × 10^6^, 3.33 × 10^6^ and 1.67 × 10^6^ cells/mL with a constant volume of 150 µL. The cell concentration was reduced by increasing both the resuspension volume of CCS and therefore increasing the equal volume of 20% DMSO/CCS at the cryopreservation stage. This was performed to maintain the concentration of 10% DMSO in the cryopreservation media. The cells were stored in liquid nitrogen for 1.7 months and then thawed using the method described above. Four individual vials were thawed from the 4 different cell concentrations, i.e., the experiment was run in quadruplicates.

### 4.4. Statistical Analysis

For each response, a separate linear mixed model was fitted with fixed treatment factors: number of days blood stored before processing and temperature of storage and their interaction. Temperature at storage had two levels, i.e., 4 °C and room temperature (RT), while number of storage days had 10 levels from day 0 to day 9. To allow for variability between donor samples a random effect for donor was included in the model. Models were fitted using the package lme4 [37] available in the open-source statistical software platform R [38]. A square root, log or logit transformation of the response was considered in order to meet the underlying assumptions of normality and constant variance of residuals. Back transformation of predicted values and standard errors of transformed responses was undertaken [39]. A significance level of 5% was used for fixed effects. Testing was undertaken in a hierarchical manner with the interaction tested before main effects. The random effect for donor was left in the model unless the variance went to the boundary. 

If a significant treatment term was found, and the interest was to compare all possible pairs of means a Tukey’s test was used to adjust the level of significance to an overall level of 5%. Otherwise, where the interest is looking at specific comparisons of means determined apriori a least significant difference (LSD) test was used. For this experiment, where there is an interaction the interest is in determining whether there is significant difference between 4 °C and RT at a particular day or for a particular temperature whether storage days 1 to 4 are different from the initial processed samples (day 0). Due to the loss of samples after each day of storage, LSD were performed to see the differences in measures post-thaw and post-incubation between 4 °C and RT in day 1 to day 4 only. 

Similar linear mixed models were fitted with a single fixed treatment factor for volume which had 7 levels (500 µL, 300 µL, 200 µL,150 µL, 100 µL, 50 µL and 25 µL) and a single fixed treatment factor for cell concentration which had 4 levels (10 × 10^6^ (i.e., 1 × 10^7^) cells/mL, 6.67 × 10^6^ cells/mL, 3.33 × 10^6^ cells/mL and 1.67 × 10^6^ cells/mL). For these experiments, where the treatment factor was found to be significant, Tukey’s test was used to compare pairs of levels.

To determine if a storage volume or storage viability is associated with post-thaw and post-incubation recovery and viability, a multiple linear regression model was used to explore the associations between either volume or viability obtained at cryopreservation and either viability or recovery after thaw and after incubation. Storage temperature and number of storage days were used as confounders in this analysis. 

## 5. Conclusions

This study generated, for the first time, empirical data to support improved quantity and quality of PBMC if blood is stored at 4 °C versus RT, when processing is delayed >24 h post-venipuncture. When delays in blood processing occur;

After 24 h, samples stored at RT have higher viability and yield than samples stored at 4 °C with no statistical difference in these variables when compared to samples processed immediately. When thawed these samples have less viability and recovery after an overnight incubation when compared with samples processed immediately. Further analysis reveals that 24 h of storage reduces the relative proportions of Monocytes and NK cells and a significant decrease in IFN-γ release upon PHA stimulation with a similar trend with CEF peptide stimulated cells (*p* = 0.68).After 48 h, blood samples stored at 4 °C have >50% yield of viable cells still present. When thawed, these samples have a decreased post-thaw viability compared to samples cryopreserved without delay. Samples stored at RT have an increase in contaminating LD neutrophils coupled with an increase in T cells and a decrease in B cells compared with samples stored at 4 °C. Interestingly, IFN-γ release has returned to similar levels of samples processed without delay.After 3 days, blood samples at 4 °C have higher yields than samples stored at RT.When these cells are thawed they have a decrease in post-thaw recovery.After 4 days, four out of five (80%) of the samples store at RT are completely lost whereas the samples stored at 4 °C still have similar viability to samples stored for 3 days.>4 days, 100% of samples stored at RT are completely lost. Samples stored at 4 °C still have ≈40% viability until day 8. Decreasing yield/day is observed until some samples are completely lost at day 9.This finding supports the proposal that blood can still be collected and processed for viable PBMC from participants enrolled in clinical studies living in remote regions that might experience transportation delays.Additionally, our data suggest that PBMC can be successfully stored at ≥6.67 × 10^6^ cells/mL in ≥150 µL without significant loss compared with a sample five times the size. This finding provides evidence for more opportunities to store PBMC from infants and children, as well as from individuals with limited blood draw volumes (<1 mL).

## Figures and Tables

**Figure 1 ijms-22-09129-f001:**
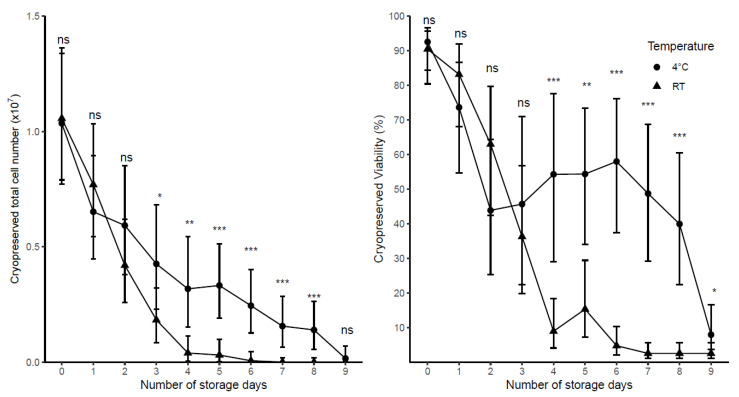
The impact of delaying blood processing on cryopreserved total viable cell number and viability of samples stored at either RT (n = 5 adult samples) or 4 °C (n = 5 adult samples) prior to processing. Cells were prepared for cryopreservation at a cell density of 10 × 10^6^ cells/mL. Day 0 represents the day of collection. Each point represents the predicted mean with 95% CI. Significance (LSD) is the difference between 4 °C and RT at a particular day; where ns = not significant, * *p* < 0.05, ** *p* < 0.01, *** *p* < 0.001.

**Figure 2 ijms-22-09129-f002:**
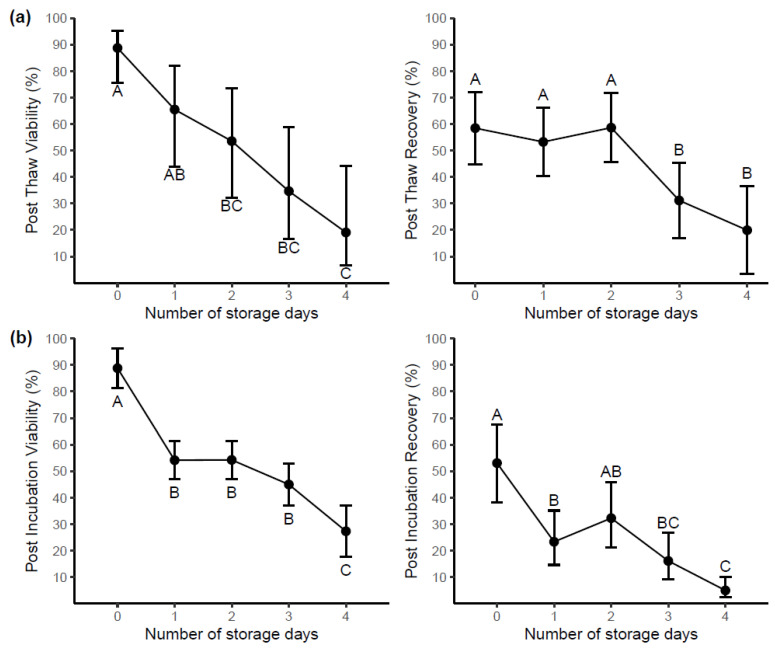
The impact of delaying blood processing on samples (n = 10). Cells were prepared for cryopreservation at a cell density of 10 × 10^6^ cells/mL. (**a**) Post-thaw viability and recovery of samples stored for up to 4 days prior to processing. (**b**) Post-incubation (overnight at 37 °C) viability and recovery of samples stored for up to 4 days prior to processing. Day 0 represents the day of collection. Each point represents the predicted mean with 95% CI. Letters (A, B, C) indicate groups (Tukey’s test), where predicted means that are not significantly different are labeled with the same letter. No significant differences between 4 °C and RT were found, therefore storage day means only are shown.

**Figure 3 ijms-22-09129-f003:**
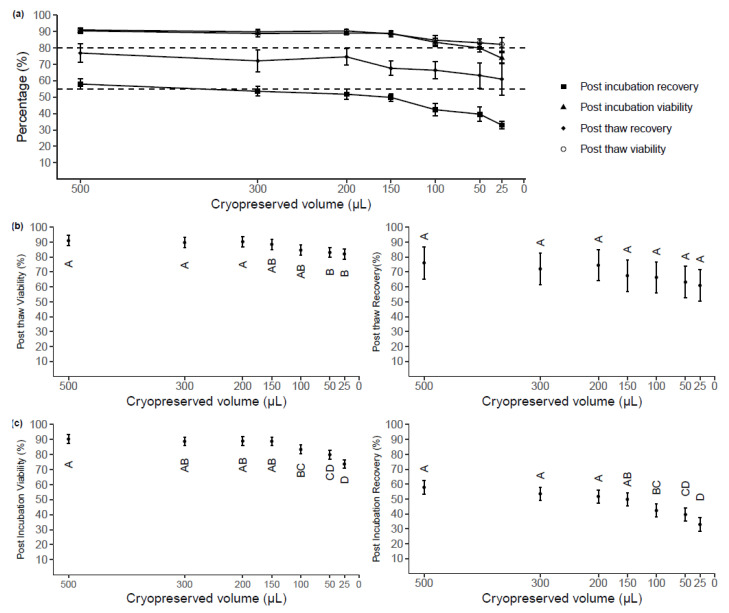
(**a**) Post-thaw and post-incubation viability and recovery from samples stored at a cell density of 10 × 10^6^ cells/mL at seven different volumes. Each point represents the mean of raw data with SD (n = 3). Acceptable criteria: Post-thaw viability ≥80%, post-thaw recovery ≥55% (**b**) The impacts of cryoprotectant media volume on post-thaw viability (predicted mean with 95% CI, n = 3). (**c**) The impacts of cryoprotectant media volume on viability measured post-overnight incubation at 37 °C (predicted mean with 95% CI, n = 3). Letters (A, B, C, D) indicate groups (Tukey’s test), where predicted means that are not significantly different are labeled with the same letter.

**Figure 4 ijms-22-09129-f004:**
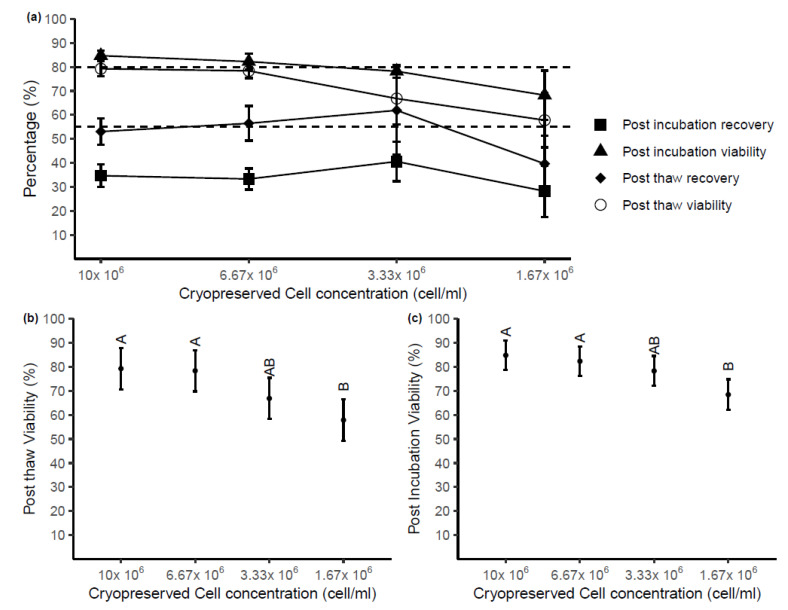
(**a**) Post-thaw and post-incubation viability and recovery from samples stored at decreasing cell concentrations in 150 µL of cryopreservation media. Each point represents the mean of raw data with SD (n = 4). Acceptable criteria: Post-thaw viability ≥80%, post-thaw recovery ≥55% (**b**) The impact of cryopreserved cell concentration on post-thaw viability. Means that are not significantly different in statistical analyses are labeled with the same letter (predicted mean with 95% CI, n = 4). (**c**) The impact of cryopreserved cell concentration on viability and after overnight incubation at 37 °C (predicted mean with 95% CI, n = 4). Letters (A, B) indicate groups (Tukey’s test), where predicted means that are not significantly are labeled with the same letter.

**Table 1 ijms-22-09129-t001:** The comparison in cryopreserved viability and total viable cell number measured at different days of storage in either 4 °C samples or RT samples.

Day	Cryopreserved Viability (%)		Cryopreserved Total Viable Cell Number (× 10^7^)	
	4 °C	RT	4 °C	RT
0	92.53 (84.30, 96.62)	90.45 (80.40, 95.62)	1.04 (0.77, 1.34)	1.06 (0.79, 1.36)
1	73.61 * (54.73, 86.55)	83.14 ns (68.11, 91.92)	0.65 * (0.45, 0.90)	0.77 ns (0.54, 1.03)
2	43.84 *** (25.28, 64.31)	62.97 ** (42.42, 79.69)	0.59 ** (0.38, 0.85)	0.42 *** (0.26. 0.62)
3	45.66 *** (22.41, 70.97)	36.25 *** (19.77, 56.76)	0.43 *** (0.23, 0.68)	0.18 *** (0.08, 0.32)
4	54.27 *** (28.97, 77.54)	8.90 *** (4.06, 18.39)	0.32 *** (0.15, 0.54)	0.04 *** (0.00, 0.11)

^(*)^ Data are predicted means (95%CI). Significance (LSD) is comparison with day 0; where ns = not significant, * *p* < 0.05, ** *p* < 0.01, *** *p* < 0.001.

## Data Availability

The data generated during the study can be found at https://adelaide.figshare.com (accessed on 9 August 2021).

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
