# Peer review of "Optimization of Blood Handling and Peripheral Blood Mononuclear Cell Cryopreservation of Low Cell Number Samples"

_ijms, 2021, doi:10.3390/ijms22179129_

Round 1

Reviewer 1 Report

The authors have made considerable helpful changes to the manuscript.  However, they have not addressed some necessary comments from the previous review.  Of note - if the authors provided a point-by-point response to review, it was not shared with reviewers.

My major concern is that providing only the predicted model mean data throughout the manuscript – rather than the actual data per subject – may obscure variability in count, viability, and immunological data.  It is not clear to me why the actual, raw data could not be provided in supplemental figures, with individual points for each subject rather than the box plots.  Even the data in most tables are predicted means rather than actual values. This should be corrected prior to acceptance of this manuscript.

The new immune assay data are very welcome.  Certain data are described but not shown – including the CEF stimulated ELISPOT counts (line 235) – where a significant reduction in spot count of 933 spots was observed.  Such a dramatic change in counts should be discussed much more fully as a major caveat of delay-to-processing – if a 70% reduction in spot counts were observed with islet peptides after only 24 hour delay-to-processing, this would likely result in an inability to detect any islet-specific responses above background given the known, lower magnitude of response to islet peptides/protein compared to CEF or PHA.  This should be clearly incorporated in the discussion, which currently reads as if there are few to no concerns with the processing delay.  Indeed, the discussion states there is no change after 2 days of delay-to-processing (line 379) – this is in dramatic contrast to the data on line 235.  Perhaps it is attributable to low sample availability at day 2? How there be no significant difference from day 0 otherwise?  If attributable to low sample availability, this is probably not the best datapoint to focus upon in the discussion - and the day 1 data should be more fully considered.  In addition, the raw data for lines 457-462 must be shown in order to justify any claim that there is no concern with delay-to-processing for CEF stimulation.

Minor – B cell data are not shown (line 269).  Why?

Minor – The supplemental figures need legends. 

Minor – Supplemental Figure 2 – Y axis goes to -1000 spots for the PHA data. Why would this be? Please provide raw data for CEF and PHA rather than models.

Author Response

Please see the attachment. Thank you very much.

Best regards,

Dao.

---------

RESPONSES TO REVIEWER 1 COMMENTS

  1. The authors have made considerable helpful changes to the manuscript.  However, they have not addressed some necessary comments from the previous review.  Of note - if the authors provided a point-by-point response to review, it was not shared with reviewers.

Response

We thank for your comments and suggestions. We have addressed the issues and separately responded to each reviewer.

  1. My major concern is that providing only the predicted model mean data throughout the manuscript – rather than the actual data per subject – may obscure variability in count, viability, and immunological data.  It is not clear to me why the actual, raw data could not be provided in supplemental figures, with individual points for each subject rather than the box plots.  Even the data in most tables are predicted means rather than actual values. This should be corrected prior to acceptance of this manuscript.

Response

We thank the reviewer for their comment. While not showing the actual raw data points, the current plots of predicted means more than adequately capture the variability present in the raw data by including the 95% confidence intervals around the predicted means. We took on board both the reviewers comment changing the standard errors on the graphs to the 95% confidence interval which reflect the variability in the predicted mean that one might expect on 95/100 occasions if we repeated the experiment 100 times. The plots presented in the results section reflect the results of the analysis and support the accompanying discussing in the text. Figures 3a and 4a did show the means and standard errors of the raw data but not the actual raw data points. While we are happy to present the raw data plots in supplementary material (Figure S4 – S7), we believe there is little information to be gained from this, given we will also be making the raw data itself available to readers (at Figshare- see below), where they can if they choose to explore the data further for themselves.

https://adelaide.figshare.com/articles/dataset/Dataset_Experiment_1_Figure_1_Impact_of_delaying_blood_processing_on_cryopreserved_total_viable_cell_number_and_viability/15132666

https://adelaide.figshare.com/articles/dataset/Dataset_Experiment_1_Figure_2_Impact_of_delaying_blood_processing_on_post_thaw_and_post_incubation_parameters/15132678

https://adelaide.figshare.com/articles/dataset/Dataset_Experiment_1_Functional_analysis_and_Immunophenotyping_analysis/15132684

https://adelaide.figshare.com/articles/dataset/Dataset_Experiment_2_Impact_of_cell_volume_on_PBMC_quality/15132708

https://adelaide.figshare.com/articles/dataset/Dataset_Experiment_3_Impact_of_cell_concentration_on_PBMC_quality/15132714

  1. The new immune assay data are very welcome.  Certain data are described but not shown – including the CEF stimulated ELISPOT counts (line 235) – where a significant reduction in spot count of 933 spots was observed.  Such a dramatic change in counts should be discussed much more fully as a major caveat of delay-to-processing – if a 70% reduction in spot counts were observed with islet peptides after only 24 hour delay-to-processing, this would likely result in an inability to detect any islet-specific responses above background given the known, lower magnitude of response to islet peptides/protein compared to CEF or PHA.  This should be clearly incorporated in the discussion, which currently reads as if there are few to no concerns with the processing delay.  Indeed, the discussion states there is no change after 2 days of delay-to-processing (line 379) – this is in dramatic contrast to the data on line 235.  Perhaps it is attributable to low sample availability at day 2? How there be no significant difference from day 0 otherwise?  If attributable to low sample availability, this is probably not the best datapoint to focus upon in the discussion - and the day 1 data should be more fully considered.  In addition, the raw data for lines 457-462 must be shown in order to justify any claim that there is no concern with delay-to-processing for CEF stimulation.

Response

We wish to thank the reviewer for their question. The statistical modelling indicates that delay to processing and storage temperature alone or in combination did not impact the CEF counts with a p-value of 0.068 (for delay to processing). We have altered this and re-iterated the sampling limitation to this result in the results section (line 169 – 170). Furthermore, we have explicitly mentioned mean values in text and supply the figure in supplementary (Figure S5).

“…Of note, the impact of storage days prior to processing p-value was equal to 0.068, due to the limited number of RT day 2 samples (n=1)…”

We did not observe a decrease of 933 or 70% of CEF stimulated spots and we have corrected this as it was an observation we had with PHA stimulated cells, this is now reflected in the results (from line 173 to 175)

“…However, the count significantly reduced by 933 spots (~70%) when the samples were stored 24 hours prior to processing, regardless of storage temperature (Supp. Figure 1)…”

In the discussion we originally had a statement that day has no impact on CEF, we have now re-structure the sentence around the observations. i.e. we saw a drop and rise again (from line 296 to 299).

“…Lastly, samples stored with a delayed of 2 days at 4°C did not have significantly different CEF or PHA stimulated IFN-γ release compared with day 0 baseline samples, however samples stored for 1 day had a  decreased PHA stimulated IFN-γ release…”

  1. Minor – B cell data are not shown (line 269).  Why?

Response

The line 269 described the results of experiment 3 that the PBMC quality evaluated based on viability and recovery measured post thaw and post incubation.  No functional analysis regarding B-cell data existed in this experiment. 

However, data points of B cell analysis in experiment 1 has been added into Figure S5 and raw data is available at Figshare

https://adelaide.figshare.com/articles/dataset/Dataset_Experiment_1_Functional_analysis_and_Immunophenotyping_analysis/15132684

  1. Minor – The supplemental figures need legends. 

Response

The Figure S2 has been divided into 2 separated Figure S2 and S3. The legend has been added into each figure.

  1. Minor – Supplemental Figure 2 – Y axis goes to -1000 spots for the PHA data. Why would this be? Please provide raw data for CEF and PHA rather than models.

Response

Figure S2 regarding PHA analysis has been fixed. All CEF and PHA data points were also presented in Figure S5

Reviewer 2 Report

Dear Authors, 

with this work you are interestingly dissecting issues related to PBMC management, suggesting the best protocol to use when considering a delay in the processing of this difficult sample. The results are finely presented, with a methodical and precise description of data you reported. It is clear that you are describing an observed phenomenon like degradation of PBMC during time in some conditions with the precise aim to overcome this effect, proposing practical and efficient solutions.

Although the manuscript is well structured, I would suggest minor revisions to be considered for publication. 

(1) Titles of results paragraphs should have the same type: bold or italic

(2) Figure 2 need to be fix well: (a), (b) should be palced at the top left of the Figure

Line 94: change effecting with affecting 

Line 95-97: the sentence is not clear, rearrange

Line 104: of processing

Line 294: why 4,4 months?

Line623: equation (2) should be written better, because is not allineated in all its terms

Author Response

Please see the attachment. Thank you very much.

Best regards,

Dao.

----------

RESPONSES TO REVIEW 2 COMMENTS

Dear Reviewer,

We thank for your suggestion and comments. We have addressed the issues and reponded as following.

  1. Titles of results paragraphs should have the same type: bold or italic

Response

Yes, titles of result paragraphs have been re-formatted as suggested.

  1. Figure 2 need to be fix well: (a), (b) should be placed at the top left of the Figure

Response

Figure 2, Figure 3 and Figure 4 have been re-formatted i.e. (a), (b) and (c) letters were placed at the top left of each Figure.

  1. Line 94: change effecting with affecting 

Response

The typo has been fixed in line 83 as following:

“…To our knowledge, there are no studies that have been conducted to explore a low volume cut-off, where the PBMC are stored without affecting cell viability and recovery post-thaw…”

  1. Line 95-97: the sentence is not clear, rearrange

Response

The sentence was re-written in line 94 – 96 as following

“...We also reveal optimized storage conditions for PBMC yields as low as 1x106 cells at which post thaw viability and recovery were equal to cells stored at 5x106 cells in 500µL...”

  1. Line 104: of processing

Response

The sentence was fixed in line 100 as following

Effects of storage temperature and of delayed processing on PBMC viability and recovery

  1. Line 294: why 4,4 months?

Response

After the isolation, PBMCs were stored in liquid nitrogen until thawed in batch. Therefore, the average of times from LN2 storage to thaw of all PBMC samples were 4.4 months.

  1. Line 623: equation (2) should be written better, because is not allineated in all its terms

Response

Equation (2) has been re-formatted as following (attachment)

Round 2

Reviewer 1 Report

The authors have responded adequately to my review. 

This manuscript is a resubmission of an earlier submission. The following is a list of the peer review reports and author responses from that submission.

Round 1

Reviewer 1 Report

The authors here seek to identify time and temperature conditions under which blood samples may be processed for immune assays outside the “normal” window of 24 hours.  This is a laudable goal.  In addition, the introduction notes a few of the negative consequences of delayed processing times on immune readouts.  However, subsequent statements throughout the manuscript seem to downplay the considerable problems that occur with delayed sample processing.  For instance, in figure 1 there is no significant difference between RT and 4C processing at 48 hours after collection -  but the considerable, biologically meaningful difference from baseline for both temperature groups is underdiscussed.  Similarly, the discussion opens with the statement that “storing at 4C can maintain the viability and total cell number when storing at RT > 24 hours cannot.”  This is not supported by the data in Figure 1; both temperatures show considerable declines over time, it’s just that they are similar in the early timepoints after collection.  Statements like these should be corrected throughout the manuscript, including statements in the conclusion at lines 455-457.  Further discussion of the expected impacts of low cell recovery should be described, especially given that loss of specific cell populations may be biased to specific immune cell populations, further biasing downstream results.  If the ENDIA study will be running downstream assays that are not expected to be impacted by delays to processing, this should be noted, and any data indicating that delays to processing do not impact downstream readouts should be provided.  In general, the risk here is that future readers will misinterpret your data to indicate that processing later is an “ok” thing to do, rather than something only to do if you absolutely must and if your downstream assays are compatible – I don’t doubt that the authors would want to avoid this misinterpretation as well, but tweaks to the wording throughout will help avoid this possibility.

Throughout the manuscript, it would help to have the ranges of each measure presented; the mean/SEM “plungers” make it difficult to see the variability between individuals.  Especially for Figure 1 where only 5 individuals per timepoint (and no replicates) are included, it should be feasible to just show every actual value.  For the later figures, a confidence interval or SD would help to better understand the actual range of values that would be anticipated in the real world, since it is unlikely that there is no variation between people/blood draws for these measures. Knowing the frequency of acceptable values per timepoint/volume/etc would aide in interpretation of the findings.

(Minor) Figure 3 shows overnight incubation volumes down to 25 uL.  I did not see the size of plates in which these were incubated, but in many cases this would be too low of a volume regardless of the incoming viability/stability of the cells.

(Minor) Figure numbers are mislabeled at multiple points throughout the manuscript.

(Minor) Was the incubation temperature overnight really 56C (line 386)? This seems extremely high and could have had its own impact on viability.

(Minor) Why was RBC lysis only performed on later timepoints in Experiment 1? Could this have impacted your results?

Reviewer 2 Report

The study is well conducted and supported by detailed methods and materials, and very well explained. A limitation is the low number of samples, but the proof of principle of the study is solid.

No major observation to the study is needed.

Row 453: eliminate the comma 

Reviewer 3 Report

General The impact of delays in PBMC processing and the problems of dealing with very small blood volumes are real-world problems that deserve careful study and this paper has some interesting data that could be helpful. However this needs some work and possibly the addition of some functional data before publishing. General summary of most important points • The presentation needs to be simplified and shortened- terms like ‘cryopreserved cell counts’ should be replaced with ‘cell yield before cryopreservation’ and ‘cryopreserved viability’ with viability. (this led to much confusion in the first part of the paper!) The narrative is unnecessarily repetitive and could be condensed into something much smaller and less confusing; phrases like ‘cells were prepared for cryogenic preservation at a cell density of 10 x 10e6’ are needlessly complicated, should be ‘cells were cryopreserved at a density of…’. The graphs are also repetitive and yet differently presented for no discernible reason, see specific comments- tables 1-3 could also be combined, in its current form it is too little data stretched over too much space. • It is interesting that storing at 4degC for 2 days or longer increases the pre-cryopreservation viability of isolated PBMCs compared to storing at RT for the same amount of time but as the purpose of PBMC isolation is to freeze down and batch-test the thawed samples, and the thawed samples showed no differences, the usefulness of this data is questionable. Also, although the viability is better, it is still really low compared to immediately isolated samples and the post thaw viability is way below their acceptability criteria of 80%; it is uncertain if these samples would be of any practical use. If the authors could show some sort of functional data from these 2-day samples, it would make this finding more interesting. Importantly, the discussion overstates the conclusions of the experimental data (see specific points for page 9) • The cell suspension volume data is difficult to interpret. There are so many variables that are not considered including 1) the size of the cryovial relative to the amount of liquid being stored- would the findings be similar if they used a 0.5 ml vial instead of a 2 ml vial? 2) the length of 37degC incubation (which they changed according to volume) 3) the rapidity of dilution during thaw (much higher when starting with a small volume of cells), 4) the impact of different cell densities during the overnight incubation – would they have seen the same effect if all densities had been adjusted to 2M/ml? Also important details were left out of the methods- “the cells were plated” how, in what? • However, the post-thaw viability data is useful as it clearly shows the impact of volume and freezing density at the time of thaw (without compounding the results by overnight incubation at different densities) and the recommendation of not storing <150 ul cells in 2 ml cryovials and at higher than 6 M cells/ml seems logical. It would be nice to see the individual data points in each category. The data are based on 4 samples each measured once, which isn’t very robust. And if I am reading the graphs correctly, there is no significant difference in figure 4 between 7 M/ml and 3 M/ml, and the only significant difference (both at thaw and post incubation) is at 2M/ml. Shouldn’t the recommendation be to be equal to or greater than 3.33 M/ml? • When comparing the differences between two groups both means should be stated with 95%CI instead of just subtracting and providing one number. The graphs are shown as average +/- SEM and while not actually wrong, this is misleading at this sample size as it artificially minimizes the variability between the samples. It is dangerous to use parametric tests with such a small n. And Fisher’s LSD test is usually no longer recommended for post-hoc tests as it does not correct for multiple testing and thus overestimates the significance. Specific Page 1 • 24 the phrase ‘to optimize protocols’ does not make sense in this sentence, is this a mistake? • 31 the fact that it was not different than an unspecified control sample is not informative? Page 2 • 48 pediatric samples are not different than blood? • 84 the Higdon paper referred to looked at cell densities as low as 2 x 10e6 cells/ml and found no differences in post thaw viability • 85/86 cell concentration and total volume of stored sample are two separate issues, this sentence is running both concepts together. Low blood volume is sufficient description (the links to taking samples from small children has already been pointed out) • 89 the narrative would be simplified if the authors referred to ‘storing low volume blood samples’ • 90/91 the text is unnecessarily repetitive, the previous sentence already mentioned the problem of extended delays. This is one of many such examples. • 93: are both the 5 M/ml and the 1 M/ml stored in total volumes of 500 ul? • 94: will improve isolation and recovery of PBMCs from small-volume blood draws (0.5-1 ml) Page 3: • 105 by viable cell number do the authors mean ‘recovery’? the axis labelling with ‘cryopreserved’ is unclear- does this mean viable cells recovered after thaw, or simply the yield of cells isolated? Similarly, what does ‘cryopreserved viability’ mean? Is this the viability before or after cryopreservation? Clarify that the ‘storage days’ refer to ‘days between blood draw and PBMC isolation” otherwise it may mean ‘storage days in freezer before thawing’. • Figure 2: The figure should be labeled A and B. Is the difference between 0 and 1 significant (certainly looks as if this is so), in which case it should be mentioned at least briefly. Or the letter method of denoting significance that is employed in later figures also added. • 114. “The number of days which blood was stored only impacted viable cell number after 3-4 days of storage (p<0.05)”. Are the authors referring to viability or total cell number (I assume they are only counting viable cells as part of the yield)? Mean cell numbers decrease by about 50% between 0 and day 2 so how does this not impact the cell number? SEM is not a very accurate indication of error when the sample size is so low, SD would be a more reliable indicator of how spread out the values are. Also when the authors say ‘significantly impacted’ it would be good to have numbers to indicate how different the values are (with 95% CIs). • 121 how are the authors arriving at these numbers? Table 1: data represented as differences in what? Please provide more details. Indicate the test in caption. Again, with these low numbers it would be better to show the 95% CI than the SEM- the data look misleadingly tight as shown by the discrepancy between graph and conclusions in the text (eg. no difference in 0 and 1 day in yield/viability). Page 4 • 146- what are ‘post-incubation measures’? If you mean viability and recovery just say so • 148-153 – This is explained in a very complicated way.. the English needs a bit of work. Consider something like this: In contrast to fresh samples, thawed samples showed no significant differences in either viability or recovery between samples stored at RT vs. samples stored at 4degC. However, after 1 day of incubation post thaw, both viability and recovery were significantly lower in samples with processing delays compared to samples that were processed within 6 hours of blood draw (provide means with 95%CI here and refer to table, no need to go through all the table numbers (which are the same as the graphs) in such detail). • Figure 2 Caption- jumbled, pls review. Prior to processing- be specific. Prior to PBMC isolation and cryopreservation (thawing is also processing). Also, Fisher’s LSD test is no longer recommended as it does not correct for multiple testing (I got this from HJ Motulsky’s Statistics Guide for Prism 5, 2007) and will thus overestimate differences. Tukey’s or Bonferroni’s are more suitable post-hoc tests according to Prism. Page 5 • 163 Again, I am not sure it makes sense to just report the differences of the mean without a 95% CI. It is misrepresenting the power of the data. • I think one big table with a series of rows for pre-cyropreservation, post thaw and post 1 day incubation would be much clearer. Table 3 header is jumbled. There is a lot of repetition in this way of showing the data • 180 I do not understand how the authors adjusted for storage temperature and days of processing delays? Surely cells that are less healthy (eg. ones that are stored in low oxygen surrounded by agitated grans for > 6 hours) will reveal increased deficits in survival after stress (eg. freeze-thawing)? How does it follow that the lower cell concentration or amount of media was responsible? These are reasonable hypothesis but I don’t see how it can be concluded that one of these two factors is the cause? • 182 “This indicated that an increase in cell concentration, and therefore cryoprotective media, may increase the viability of cells post-thaw” Another example of foggy language. I don’t understand ‘and therefore cryoprotective media’. And therefore what of the cryoprotective medium? It increased? Also, what quality of the cryoprotective medium are they referring to? Volume? If the concentration was constant, the medium volume would change. If the concentration increased, why would the medium volume also increase? I do not understand how the two parameters are related in the above sentence. Please clarify. Page 6 • 198 Figure 3a! • 203 part of the sentence is missing. • The entire description of figure 3 could be more concise. • The effect of freezing very small volumes in a large cryovial is not taken into account here. • 215 I’m not sure what ‘this is considered a biologically significant decrease’ means, why? If it is significantly different according to logical statistical tests then it is different. • 216 indicating that 100 μL is not sufficient to protect PBMC from dying after cryopreservation and an overnight incubation at that cell concentration. I would suggest ‘That a volume of 100 ul is too small for optimal cryopreservation’- I think you would find that the viability would be different at thaw if you were to use a less blunt instrument than trypan blue (eg. a marker of early apoptosis). Although the point is well taken that the incubation density could impact the result as well.. something that was not corrected for in this experiment. • Figure 3, again label different plots with A, B, C.. • 3A is confusing, what is the purpose of showing recovery and viability on the same plot suddenly? And aren’t the latter four plots the same information? It seems unnecessarily stretching the information and adds nothing • The authors should show the individual points for each volume instead of the mean + SEM. Especially if they are going to separate all four sets of data this information can be shown. Page 7 • 233 Fig 4a not 3a • It is confusing that the post incubation values are mentioned in this section before the post thaw values. • Why would the same format not be used as in figure 3? Where is the recovery suddenly? And again the two dotted lines for recovery and viability cut off are identical and confusing, indicating a range. Also aagain dvise separating into four plots with the individual data points shown instead of mean and SEM. Page 8 • 256 “We show that storing at 4oC can maintain the viability and total cell number when storing at RT >24 hours can not.” The authors do show that storing at 4degC can maintain a BETTER viability and yield compared to storing at RT for the same (extended) length of time but do not show that there is no difference of either parameter compared to the day 0 or even the day 24 values- so viability and yield is not maintained by storing at 4degC. The text describing the results is quite convoluted but I believe this is what they are saying. So this statement is misleading? Page 9 • 284 “Our results showed that storage at 4oC also maintains the post thaw viability when blood needs to be stored/transported longer than 2 days after venipuncture. The same results were also found in all measures in the blood stored for 3 days. Therefore, this is the first study, to our knowledge, reporting that the blood stored at 4ï‚°C results in the higher recovery of cells when the time interval between blood withdrawal and PBMC isolation exceeded 24 2 hours, compared with blood stored at RT”. I thought figure 2 showed that there was no significant difference between storage at RT and 4degC when looking at any of the post thaw parameters? What we do see is viability before cryopreservation being higher is that what is meant? This is misleading when following the previous statement which talks about post thaw viability. The authors show no difference of storage temperature after thawing so what is the point of having higher yields post PBMC isolation before cryopreservation? This may have an impact on assays done on whole blood or extracted PBMCs but I think PBMCs are rarely extracted and used without cryopreservation in clinical studies. • Also if going by their criteria of acceptability- the samples which were preserved at 4degC for 2 or more days prior to PBMC isolation before freezing do not meet this; figure 2 shows an average of 60% post thaw viability. • 300 >150 ul may well depend on the kind of cryovial used